# Illnesses and Symptoms in Older Adults at the End of Life at Different Places of Death in Korea

**DOI:** 10.3390/ijerph19073924

**Published:** 2022-03-25

**Authors:** Su Hyun Kim

**Affiliations:** Research Institute of Nursing Science, College of Nursing, Kyungpook National University, Gukchaebosang-ro 680, Daegu 41944, Korea; suhyun_kim@knu.ac.kr

**Keywords:** chronic disease, death, nursing homes, signs and symptoms, terminal care

## Abstract

Context: A comprehensive plan has been launched by the Korean government to expand hospice and palliative care from hospital-based inpatient units to other services, such as palliative care at home, palliative consultation, and palliative care at a nursing home. Objective: To examine the illnesses and symptoms at the end of life associated with the place of death among older Korean adults. Methods: This secondary data analysis included a stratified random sample of 281 adults identified from the exit survey of the Korean Longitudinal Study of Aging aged ≥65 years and who died in 2017–2018. Results: Overall, 69% of the patients died at hospitals, 13% died at long-term care facilities (LTCF), and 18% died at home. In the multinomial logistic regression analysis adjusting for age, sex, and marital status, older adults who died in the hospital had higher odds (2.02–4.43 times) of having limitations in activities of daily living (ADL) as well as symptoms of anorexia, depression, weakness, dyspnea, and periodic confusion 1 month before death than those who died at home. Older adults who died in an LTCF were more likely to have limitations in ADL and instrumental ADL as well as a higher likelihood (2–5 times) of experiencing pain, anorexia, fatigue, depression, weakness, dyspnea, incontinence, periodic confusion, and loss of consciousness than those who died at home. Conclusion: Since the majority of subjects died either in a hospital or an LCTF, and this proportion is expected to increase, policy planning should focus on improving the palliative case in these settings. Future policies and clinical practices should consider the illness and symptoms of older patients at the end of life across different care settings.

## 1. Introduction

The number of older adults dying of chronic illnesses, such as cardiovascular diseases, chronic obstructive pulmonary disease, diabetes, cancer, and dementia across long-term disease trajectories, is increasing worldwide [1]. Accordingly, these people require ample palliative care that covers various debilitating symptoms presenting at the end of life. However, poor palliation of symptoms and concerns with the quality of care at the end of life are often reported in the current literature [2]. Therefore, to ensure adequate end-of-life care, patient-tailored palliative care must be provided at each care setting for older adults.

Although most people wish to spend their last moments at home, hospitals remain a common place of death in many countries [3]. Recent reports report that 76.2% of all deaths in South Korea [4], >80% of deaths in Japan [5], and 73% of deaths in Turkey [6] occurred in hospitals. Notably, this high number of hospital deaths in Korea has been attributed to the unavailability of competent palliative care services for the end of life outside hospitals [7]. In 2015, the Korean government started a national health insurance coverage for hospice and palliative care, but it was limited to inpatient services in the hospital units for patients with terminal cancer [8]. Eventually, the Korean government launched a comprehensive plan in 2019 to expand hospice and palliative care for patients with other serious multiple illnesses and introduced other services, such as palliative care at home, palliative consultation, and palliative care at nursing homes [7].

To develop successful palliative care services for older adults in diverse settings, it is essential to identify the common illnesses and symptoms experienced by these patients at the end of life in these care settings. In terms of place of care in Korea, hospitals have remained the most common place of death over the last several decades, with an increasing proportion of patients dying at long-term care facilities (LTCFs). Conversely, the proportion of patients dying at home is decreasing [7].

According to a conceptual model by Gomes and Higgins in 2006, the place of death is significantly associated with illness factors, besides the patients’ demographic and environmental factors [9,10]. The illness factors include clinical changes that occur because of illness, such as pathological changes during the illness, functional status, and symptoms [9,10]. Older adults often have multiple comorbid chronic illnesses, which lead to unmanageable symptoms at the end of life, including pain, dyspnea, constipation, urinary incontinence, restlessness, confusion, nutritional problems, and weakness, necessitating serious attention from palliative care services [11]. As demographic and environmental factors are mostly nonmodifiable, characterizing these illnesses and their symptoms in older adults at the end of life in each care setting may provide meaningful information to clinicians involved in the development of palliative care policies and programs.

The existing literature describes inconsistent findings regarding the illness factors associated with the place of death. In a systematic review about terminally ill patients with cancer, death at home was associated with a long duration of disease and low functional status, while disease symptoms and pain were not associated with the place of death [10]. Another study reported that higher functional status and greater pain intensity predicted institutionalization as the final place of care in older adults with cancer enrolled in home-based palliative care service [12]. A study involving geriatric outpatients in Turkey stated that the prevalence of death was higher for in-hospital older patients with chronic renal failure, but no other comorbid condition was associated with the place of death [6].

Although most older adults suffer from multiple illnesses and symptoms at the end of life, there is limited data regarding the prevalence of symptoms and illnesses in older adults with multiple comorbidities at death in common care settings [6,9]. Previous studies analyzing the determinants for the place of death among older adults have mostly been conducted for single diseases, such as cancer [12,13,14], dementia [15,16,17], and lung, kidney, or liver diseases [18,19,20], with most data available as demographic and clinical characteristics. As palliative services need to be tailored as per the patient’s illness and symptoms, which might differ from each place of end-of-life care, more information is needed to identify the reported illnesses and symptoms at various places of care for older adults at the end of life in Korea. We conducted this study to examine the likelihood of illnesses and symptoms experienced by older adults in Korea who died at hospitals or LTCFs compared to their homes. We believe this information will help direct the expansion of palliative care services in Korea.

## 2. Methods

### 2.1. Sample and Data

Secondary data analysis was performed using the data obtained from the Wave 7 exit survey of the Korean Longitudinal Study of Aging (KLoSA), collected after the death of the subjects in the study cohort; the subjects were selected via stratified random sampling from the national census and followed every 2 years since 2005 [21]. The KLoSA exit survey was conducted as an interview with surrogate respondents of the decedents, such as family, relatives, or neighbors. Among the 551 respondents who completed the exit survey, 353 respondents related to those who died at age ≥65 years in 2017 and 2018 were selected for this study. After excluding those who died from accidents (*n* = 55), those who received hospice care service at any time before death (*n* = 14), or were uncertain of this (*n* = 3), 281 decedents were included in this study. The decedents who received hospice care at any time before death were excluded from this analysis because hospice care service was discontinued at the time of death. The institutional review board of the Kyungpook National University approved this study (approval No.: 2020-0116).

### 2.2. Measurement

We assessed the decedents’ place of death from the interviews with surrogate respondents with the question “Where did the decedent die?”. The answers were categorized into their own homes or their offspring/relatives, hospitals, and LTCFs [21]. The respondents were asked questions about the presence of chronic illnesses, such as hypertension, diabetes, cancer, heart disease, osteoarthritis, dementia, cerebral infarction, lung disease, depression, and fracture. They were also asked about the activities of daily living (ADL), instrumental ADL (IADL), presence of pain for 1 year before death, and the presence of anorexia, fatigue, depression, weakness, dyspnea, incontinence, periodic confusion, loss of consciousness, coughing, vomiting, and emotional disturbances 1 month before death using questionnaires [21]. The possible range of ADL and IADL scores were 0–7 and 0–10, respectively, with higher scores indicating a greater need for assistance [21].

### 2.3. Data Analysis

Descriptive statistics were used to summarize the data. We used the chi-square test, analysis of variance, and multinomial logistic regression analysis to examine the factors associated with the place of death. The category of home death was used as a reference to identify the odds of hospital death and LTCF death. The demographic variables, which were significantly different among the places of death, such as age, gender, and marital status, were adjusted as covariates for the multinomial logistic regression. SPSS version 25.0 (IBM Corp., Armonk, NY, USA) was used for analysis.

## 3. Results

### 3.1. Demographic Characteristics of the Sample

Among the 281 decedents, 194 (69.0%) died in a hospital, 37 (13.2%) died at LTCFs, and 50 (17.8%) died at home. At the time of death, the decedents had a mean age of 81.61 ± 7.73 years: 53.4% were female and 66.8% were married. About 68.7% had education less than elementary school and 89.3% had national health insurance. Before their death, 42.8% lived in metropolitan areas, 30.3% lived in rural areas, and 26.9% lived in small cities.

On comparing the demographic characteristics, we observed that those who died in LTCFs had a significantly higher mean age than the patients who died in hospitals or at home (F = 4.87, *p* = 0.008). Males and married subjects had a significantly higher proportion of death at hospitals and a lower proportion of death at LTCFs than females and unmarried subjects, respectively (χ^2^ = 6.90, *p* = 0.031; χ^2^ = 7.30, *p* = 0.027, respectively). No other differences were observed in the demographic variables with respect to the place of death.

Overall, the most common chronic illnesses were hypertension, diabetes mellitus, and cancer (Table 1). The mean ADL and IADL scores 3 months before death were 4.23 ± 3.21 and 5.86 ± 4.41, respectively. The common symptoms experienced 1 month before death were anorexia (69.8%), fatigue (59.8%), depression (51.6%), dyspnea (49.5%), and weakness in extremities (48.8%).

### 3.2. Illness and Symptoms Associated with the Place of Death

After adjusting for individual demographic factors regarding the illnesses and symptoms associated with the place of death (Table 1), we found that older adults who died at hospitals had lower odds (0.45 times) of having heart disease and higher odds (1.14 times) of having ADL limitations than those who died at home. Furthermore, subjects who died at hospitals had higher odds (2.02–4.43 times) of experiencing anorexia, depression, weakness/paralysis, dyspnea, and periodic confusion than those who died at home; no other symptoms had comparable results. The presence of other illnesses, including hypertension, diabetes mellitus, cancer, osteoarthritis, dementia, cerebral infarction, lung disease, depression, and fracture, was not significantly associated with increased odds of death at hospitals compared to deaths at home. Similarly, IADL score and pain intensity before death were not significantly associated with increased odds of death at hospitals.

Conversely, older adults who died at LTCFs had higher odds (2.75 times) of suffering from hypertension and experiencing ADL (1.55 times) and IADL limitations (1.40 times) than those who died at home. The presence of other illnesses, including diabetes mellitus, cancer, heart disease, osteoarthritis, dementia, cerebral infarction, lung disease, depression, and fracture, was not significantly associated with increased odds of death at LTCFs compared to deaths at home. Furthermore, older adults who died at LTCFs had higher odds (2.67–5.47 times) of having pain, anorexia, fatigue, depression, weakness/paralysis, dyspnea, incontinence, periodic confusion, and loss of consciousness.

## 4. Discussion

Korea has a high incidence of hospital deaths (69%) but a low proportion of home deaths (17.8%). These numbers are in stark contrast to the USA, where hospital deaths constitute only 19.8% of the total deaths, and 40.1% of the deaths occurred at home [2]. These results indicate that hospitals are still the primary place for older adults to spend their last days of life in Korea; therefore, there is a need to ensure adequate palliative care for older adults in the hospital settings. Notably, the prevalence of illnesses was not significantly higher in older adults who died at hospitals than those at home, except for heart diseases, which were less common among those who died at hospitals. Instead, a previous Turkish study conducted on geriatric outpatients reported that only the presence of chronic renal failure was associated with the place of death, and no other comorbid conditions, such as heart failure, cardiovascular disease, Parkinson’s, chronic obstructive pulmonary disease, and cancer, influenced the patient’s place of death [6]. This discrepancy may be attributed to the differences in the characteristics of the study sample, i.e., a national-level randomized sample of older adults in Korea for this study [21] and geriatric outpatients in a university hospital in Turkey [6]. These findings also indicate that the type of illness occurring in older adults who died in a hospital or at home in Korea was not statistically different, except for heart disease.

However, those who died at a hospital had significantly greater odds (2 times) of experiencing anorexia, depression, weakness or paralysis, dyspnea, and confusion before death than those who died at home. Interestingly, we observed that there were no significant associations of the presence of cancer and experience of pain with deaths occurring at hospitals compared to deaths at home. Rather, older adults who died at hospitals had a higher likelihood of experiencing ADL limitation (1.14 times) and several debilitating symptoms (2.02–4.43 times) than those who died at home. This finding was not in line with the previous studies, which suggested that patients with advanced cancer who had higher functional status and greater pain intensity were more likely to be hospitalized at the end of life [12]. Our results also suggest that older adults who are in their end of life and admitted to hospitals in Korea need more attention regarding a progressive functional decline and debilitating symptoms, including anorexia, depression, weakness, dyspnea, and confusion, from comorbid chronic illnesses. We cannot rule out the possibility that severe symptoms might have caused patients to move to hospitals from their homes; nevertheless, these data indicate that the debilitating symptoms of older adults at the end of life warrant extra support from palliative services and should be managed better in hospitals. Thus, prompt consideration to develop policies and programs to improve the palliative care services is required to alleviate the debilitating symptoms in older adults with comorbidities who are at the end of their life in hospitals. This includes educating staff members about palliative care, identifying patients who need palliative care, and ensuring access to specialized palliative care teams as well as inpatient and consultation services to utilize the skills and teamwork of a multidisciplinary palliative care team [22].

Conversely, the proportion of deaths occurring in LTCFs in Korea (13.2%) was lower than that reported in the USA (24.9%) [2]; however, this proportion is expected to increase in Korea due to the growing older population receiving long-term care service insurance [23]. In our study, older adults who died at LTCFs experienced high symptom burdens at the end of life. They had significantly higher odds (2–5 times) of experiencing loss of consciousness, anorexia, fatigue, pain, depression, weakness, dyspnea, confusion, and incontinence at the end of life than those who died at home; these odds were far worse than those for older adults who died at hospitals. This finding indicates that older adults at LTCFs in Korea suffer profoundly from multiple symptoms at the end of their life, which is far more debilitating than those who die at home. This higher intensity of symptom burdens among older adults at LTCFs at the end of life offers substantial evidence about the need to advance policies and programs about palliative care services at LTCFs, which must incorporate nonmalignant debilitating chronic diseases. In Korea, healthcare professionals, such as physicians and nurses, are not required to be included among the staff of LTCFs [24]. However, our findings emphasize the urgency of symptomatic management for older adults at the end of life in LTCFs, which require the employment of skilled healthcare professionals. Various approaches for palliative care should be considered in LTCFs, such as the provision of clinical nurse specialists, the use of hospice beds, and training the nursing staff [8].

This study also highlights the notable illnesses and symptoms encountered at the end of life across different care settings in Korea, which must be accounted for during direct policymaking and program development for palliative care. The place of death has mostly been discussed as an indicator of the quality of end-of-life care, but some determinants of the place of care are not modifiable [9]. Nevertheless, this study was able to underline the various illnesses and symptom-related factors that need to be addressed to ensure the quality of end-of-life care among older adults, considering the preferred location of care by the patients and their family members. In Korea, wherein hospice and palliative services are offered by the national healthcare insurance system, it is important to analyze and adequately approach these factors across different care settings, especially when these are modifiable factors.

This study had a few limitations. We used retrospective data from a secondary source reported by surrogate respondents without including various factors that affect the place of death. Moreover, since older adults under hospice services were excluded, we could not compare our study results to that of patients under palliative care services. Since data of the original study were collected from retrospective interviews with surrogates, the possibility of recall bias on symptoms experienced by older adults before death was not ruled out. Further prospective research is needed to identify comprehensive factors regarding the place of death, including patient, environmental, and healthcare service factors [25].

## 5. Conclusions

Deaths among older adults in Korea occur mostly at hospitals, and the proportion of older adults approaching LCTFs in their end-of-life is expected to increase. Although the prevalence of illness among older adults did not differ significantly between those who died at hospitals and at LTCFs and those at home, the likelihoods of experiencing anorexia, depression, weakness/paralysis, dyspnea, and periodic confusion were higher among those who died at hospitals than those at home. Additionally, older adults who died at LTCFs experienced 2–5 times the functional limitations, pain, anorexia, fatigue, depression, weakness/paralysis, dyspnea, incontinence, periodic confusion, and loss of consciousness at the end of life than those who died at home. This study indicates that illness factors should be accounted for while developing advanced palliative care facilities in diverse care settings for end-of-life care in Korea. The findings regarding the illnesses and symptoms in patients across different places of death suggest the need to create specifically tailored services that cater to the needs of older adults at the end of life according to their place of care. This will contribute to the quality of end-of-life care among older adults. Future policies and clinical practices should consider the illness and symptom burden in this population across different care settings.

## Figures and Tables

**Table 1 ijerph-19-03924-t001:** Results of multinomial logistic regression analysis about the association of illnesses and symptoms before death with the place of death (N = 281).

Variables	Total	Hospital (*n* = 194, 69.0%)	LTCF (*n* = 37, 13.2%)	Hospital (*n* = 194, 69.0%)	LTCF (*n* = 37, 13.2%)
		vs. Home (*n* = 50, 17.8%) (ref)		vs. Home (*n* = 50, 17.8%) (ref)	
	N (%)/Mean (SD)	Unadjusted OR (95% CI)		Adjusted OR (95% CI) ^a^	
**Presence of chronic illness ^b^**					
Hypertension	92 (32.7)	1.52 (0.75, 3.11)	2.69 (1.08, 6.73) *	0.99 (0.95, 1.03)	2.75 (1.07, 7.05) *
Diabetes mellitus	65 (23.1)	1.70 (0.74, 3.83)	1.94 (0.68, 5.55)	1.69 (0.74, 3.88)	1.87 (0.65, 5.44)
Cancer	61 (22.0)	2.13 (0.90, 5.05)	0.96 (0.28, 3.30)	2.45 (0.98, 6.31)	1.47 (0.40, 5.46)
Heart disease	44 (15.7)	0.48 (0.23, 1.01)	0.25 (0.07, 0.96) *	0.45 (0.21, 0.96) *	0.27 (0.07, 1.03)
Osteoarthritis	44 (15.7)	1.65 (0.60, 4.49)	2.89 (0.88, 9.52)	1.77 (0.63, 4.94)	1.99 (0.57, 6.95)
Dementia	44 (15.7)	1.52 (0.56, 4.16)	3.81 (1.19, 12.17) *	1.65 (0.58, 4.71)	2.93 (0.85, 10.06)
Cerebral infarction	35 (12.5)	1.55 (0.51, 4.70)	3.17 (0.88, 11.49)	1.51 (0.50, 4.61)	3.55 (0.96, 13.21)
Lung disease	36 (12.8)	1.94 (0.65, 5.81)	1.39 (0.33, 5.98)	1.81 (0.59, 5.49)	1.96 (0.44, 8.74)
Depression	14 (5.0)	0.76 (0.20, 2.93)	0.90 (0.14, 5.65)	0.72 (0.19, 2.80)	1.18 (0.18, 7.79)
Fracture/traffic accident	9 (3.2)	1.56 (0.18, 13.30)	2.80 (0.24, 32.10)	1.28 (0.14, 11.41)	3.25 (0.27, 39.20)
**ADL score for 3 months prior to death ^c^**	4.23 (3.21)	1.14 (1.03, 1.26) *	1.56 (1.28, 1.90) *	1.14 (1.03,1.26) *	1.55 (1.27, 1.90) *
**IADL score for 3 months prior to death ^d^**	5.86 (4.41)	1.07 (0.99, 1.15)	1.42 (1.21, 1.66) *	1.07 (0.99, 1.15)	1.40 (1.20, 1.63) *
**Pain during 1 year prior to death ^b^**	91 (32.4)	1.97 (0.97, 4.19)	3.40 (1.32, 8.77) *	1.88 (0.87, 4.07)	4.52 (1.65, 12.36) *
**Symptoms during 1 month prior to death ^b^**					
Anorexia	196 (69.8)	2.96 (1.56, 5.61) *	4.64 (1.72, 12.52) *	2.67 (1.43, 4.98) *	3.88 (1.45, 10.08) *
Fatigue	168 (59.8)	1.61 (0.86, 3.01)	3.37 (1.33, 8.57) *	1.66 (0.88, 3.13)	3.86 (1.47, 10.10) *
Depression	145 (51.6)	1.93 (1.02, 3.67) *	4.20 (1.69, 10.45) *	2.02 (1.05, 3.90) *	4.87 (1.89, 12.57) *
Weakness/paralysis	137 (48.8)	3.09 (1.55, 6.17) *	4.68 (1.87, 11.70) *	2.93 (1.46, 5.89) *	4.66 (1.83, 11.89) *
Dyspnea	139 (49.5)	4.55 (2.20, 9.40) *	3.74 (1.48, 9.48) *	4.43 (2.13, 9.19) *	4.07 (1.58, 10.51) *
Incontinence	115 (41.1)	1.89 (0.94, 3.78)	5.93 (2.33, 15.10) *	1.80 (0.89, 3.61)	5.47 (2.11, 14.17) *
Periodic confusion	116 (41.3)	2.71 (1.31, 5.60) *	4.65 (1.83, 11.83) *	2.72 (1.31, 5.64) *	4.12 (1.59, 10.67) *
Loss of consciousness	94 (33.6)	1.84 (0.89, 3.83)	3.01 (1.19, 7.64) *	1.80 (0.86, 3.76)	2.67 (1.04, 6.89) *
Coughing	81 (28.9)	1.26 (0.61, 2.59)	1.93 (0.76, 4.88)	1.16 (0.60, 2.41)	2.47 (0.94, 6.53)
Vomiting	56 (20.0)	1.76 (0.74, 4.19)	1.19 (0.36, 3.89)	1.64 (0.68, 3.97)	1.50 (0.44, 5.12)
Emotional disturbance	44 (15.7)	0.50 (0.23, 1.14)	1.14 (0.42, 3.12)	0.48 (0.21, 1.06)	1.43 (0.51, 4.06)

* *p* < 0.05; ref, reference; ADL, activities of daily living; IADL, instrumental activities of daily living; LTCF, long-term care facility; OR, odds ratio; CI, confidence interval; ^a^ Adjusted for the age at time of death, gender, and marital status; ^b^ ref, no; ^c^ possible range of 0–7, with higher scores indicating a higher level of dependence; and ^d^ possible range of 0–10, with higher scores indicating a higher level of dependence.

## Data Availability

The datasets generated and/or analyzed during the current study are available in the Korea Longitudinal Study of Aging repository, https://survey.keis.or.kr/eng/klosa/klosa01.j (accessed on 28 July 2020).

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
