# Peer review of "Illnesses and Symptoms in Older Adults at the End of Life at Different Places of Death in Korea"

_ijerph, 2022, doi:10.3390/ijerph19073924_

Round 1

Reviewer 1 Report

This is a unique study focused on the special issue "The Lived Experience of People Living with Dementia and Caregivers". Well done to the study for their efforts in examining the illness and symptoms end-of-life among Korean adults.

However, the paper requires intensively work. Many points are unclear. The authors must be provided reasons why the study concentrates specifically on the field.

1. Intro

This should be explained why is significant? What main problems are you trying to fill the gap of knowledge? What objectives are you providing to examine? See line 23-46, page 1 to 47-69, page 2). This is not clear.

2. Literature review

My suggestions would be to have a literature review sections that will then lead into the identification of illness and symptoms. It is not clear if there is a need to even consider the indicators/variable/factors. As indicated in the paper, the illness and symptoms are to some extent correlated. The following are some of the discussion I think would have been addressed in a literature review.

The following review section:

  1. Definition and context of Korean adults
  2. Defining illness and symptoms in place of death
  3. The effects of illness and symptoms

3. Methods

It is unclear if you did not provide details of appropriate methods. Also, the authors should provide a conceptual model that clearly shows the endogenous and exogenous variables. Currently, it is difficult for readers to conceptualise how the proposed illness and symptoms are linked to adults in place of death. Unless stated otherwise, I believe this is essential is beefing up the arguments and findings.

And then you should be added:

  1. Study setting
  2. Sample and sampling approach
  3. Data collection
  4. Measure variables
  5. Data analysis

4. Results

Overall, the findings presented are interesting. However, the presentations are not clear if you did not address:

  1. Descriptive Analysis (focusing on sample demographics)
  2. Intercorrelation Metric (testing among variables indicators/items
  3. Multinomial Logistic Regression (should be clear what value means?) I found your results testing, it is too short to explain what values communicate for. Why is significant (*)? Why some variables are insignificant? Should be explained why?

5. Discussion

Your discussion is to limit debate with the main findings and other studies. Why did not discuss with:

  1. explained the main findings
  2. discussed with other studies
  3. and the discuss what your findings are contributing to exist knowledge

and then separate section:

  1. Practical implications
  2. Limitations and future study

6. Conclusion

It is limited and very brief to conclude your findings. Should be addressed as follows:

  1. Described what main findings are
  2. Summary with context of Korean adults is
  3. And then summarised with the key point of the findings on theoretical/practical contributions

7. References

Both the text and the list references are incorrect. The following in MDPI formats is required.

Author Response

Thank you so much for your review. We have revised the manuscript according to your comments. 

Reviewer 2 Report

Thank you for this opportunity for reviewing this manuscript.

This is a study to examine factors associate with older adults’ place of death and the findings of this study can contribute to policy development and elderly care service provision. As the authors mentioned in the manuscript, the retrospective interviews for surrogate respondents can be a limitation. However, the obtained data possibly indicates conditions more relevant to care givers that determine older adults’ place of death. Therefore, I perceived utilizing existing data to produce meaningful findings that can be foundations for future research in the area of elderly care could be rather a strength of the present study.

Below are my comments, which I hope are useful for the authors to improve the quality of this manuscript.

Limitation L185-186: Authors should mention the possibility of bias caused on ADL and IADL measurements from retrospective interviews with surrogates.

Palliative care L69, L144, L166: Which results supported the more needs of palliative care? Provision of home-based elderly care such as general practitioners’ house call, day care service and physical therapy rather than palliative care is likely to matter considering the presence of cancer and pain did not relate with hospital death but ADL and some symptoms prior to death did.   

Author Response

(The authors gave the same response as above.)

Round 2

Reviewer 1 Report

All suggestions are revised and read well. The contents are justified in the studies and contributed by readers of the journal.